# Ice Aprons in the Mont Blanc Massif (Western European Alps): Topographic Characteristics and Relations with Glaciers and Other Types of Perennial Surface Ice Features

**Suvrat Kaushik** [1,2,*], **Ludovic Ravanel** [1,3], **Florence Magnin** [1], **Emmanuel Trouvé** [2] **and Yajing Yan** [2]

[1] EDYTEM, University Savoie Mont-Blanc, CNRS, 73000 Chambery, France
[2] LISTIC, University Savoie Mont-Blanc, Polytech, 74944 Annecy, France
[3] Department of Geosciences, University of Oslo, Sem Sælands vei 1, 0371 Oslo, Norway
[*] Correspondence: suvrat.kaushik@univ.smb.fr

**Abstract:** Ice aprons are poorly studied and not well-defined thin ice bodies adhering to high altitude steep rock faces, but are present in most Alpine-type high mountain environments worldwide. This study aims to precisely define ice aprons based on a detailed analysis of their topographical characteristics in the Mont Blanc massif (western European Alps). For this, we accurately identified and precisely mapped 423 ice aprons using a combination of high-resolution optical satellite images from 2019. To better understand their relationship with other types of glaciers, especially the steep slope glaciers and other surface ice bodies, we built a detailed inventory at the scale of the massif that incorporates nine different types of perennial surface ice bodies. In addition, an analysis using different topographic factors helped us to better understand the preferred locations of the ice aprons. We show that they predominantly occur on west-oriented steep and topographically rugged rock slopes above the local Equilibrium Line Altitude (~3200 m a.s.l.), with concave profile curvatures around them that facilitate snow accumulation. They are also found in areas underlain by permafrost. The extensive inventory also helped us to identify different types of ice aprons based on their relationships with glaciers/ice bodies. The analysis shows that ice aprons existing at the headwall of large glaciers above a bergschrund are the most dominant ice apron type in the study area, with ~82% of the total.

**Keywords:** ice aprons; steep Alpine faces; glacier/ice bodies inventory; topographic characteristics; Mont Blanc massif

## 1. Introduction

Most Alpine research dedicated to glaciers focuses on medium- and large-sized ice bodies such as cirque and valley glaciers. However, small perennial surface-ice bodies (less than 0.5 km² in extent) account for more than ~80–90% of the total number in mid-to-low-latitude mountain ranges [1–3] These minor ice features include small glaciers on steep rock slopes: hanging glaciers, ice aprons, and glacierets [4].

Even though the total area/volume of these small ice bodies is practically negligible compared to large glaciers, they are critical in mountain environments [5–7], and their small size makes them very vulnerable to global warming [1,8–11]. [12] showed that small glaciers in Switzerland had lost around 70% of their area over the last four decades, and a considerable number of small glaciers have already disappeared throughout the last century. Monitoring the evolution of small ice features could provide glaciologists with new insight into their behavior in response to the changing climate over longer timescales. For example, their disappearance or relation to other geomorphic features such as rock glaciers can have significant implications for catchment hydrology in the future [13–15].

Although recent studies have shown the significance of small perennial snow/ice features in mountain environments, ice aprons (IAs) have been poorly studied. There was a

lack of consensus on accurately defining IAs, incorporating all aspects of their evolution and dynamics. [16] defined them as 'small accumulations of snow and ice masses that stick to the topography of a glacierized basin and are usually found above the Equilibrium Line Altitude (ELA)'. This simplistic definition generalizes IAs as simple ice or snow masses found in mountain regions. [4,17] presented relatively similar definitions. Later, [3] described IAs as 'small glaciers of irregular outline, elongated along the slope, in a mountainous terrain'. These definitions fail to accurately provide any description of the morphology and dynamics. The lack of consensus in precisely defining IAs was recently tackled by [11], who defined IAs as 'very small (typically smaller than 0.1 km$^2$ in extent) ice bodies of irregular outline, lying on slopes > 40°, regardless of whether they are thick enough to deform under their weight'. Although more extensive than the previous definitions of IAs, this definition still limits our understanding of the topographic characteristics critical for distinguishing IAs from perennial snow patches and other small features of perennial surface ice.

Over the last decades, the shrinkage and disappearance of IAs appeared to be a grave concern for mountaineers, as their presence is considered necessary for many climbs worldwide. [18] focused on the changes that have occurred to the 100 most famous mountaineering routes (as initially proposed by [19]) in the Mont Blanc massif (MBM), France. They showed that the recent evolution of IAs makes many itineraries more difficult and dangerous. [11] showed that IAs have been losing surface since the Little Ice Age (LIA), with a noticeable acceleration in the area loss since the 1990s. [20,21] surveyed rockfall events > 100 m$^3$ in the MBM since 2007 and noted an increase in these events as a result of the deglaciation of IAs following the heatwaves of 2015, 2017, 2018, 2019, and 2020. As the IAs shrink, boulders formerly embedded in the ice are released as the ice melts, resulting in a high rockfall frequency from the margins of the IAs. [22] conducted texture analysis and $^{14}$C dating from an ice core extracted in the Triangle du Tacul (3640 m a.s.l.; MBM, France) and proved that the age of the ice is older than 3 ka BP at the ice/rock interface. Several other ice cores from the Western Alps show a wide range of ages, from a few decades to thousands of years [23,24], highlighting that the ice from IAs should be among the most ancient. The loss of ice volume from IAs and their eventual disappearance in the subsequent decades should thus be regarded as a sensitive glacial and periglacial heritage loss [22].

Despite this, IAs have received very little attention, mainly because their small size leads to low implications on water resources (unlike glaciers, their melting does not significantly contribute to mountain rivers) and their non-dynamic state (low risk of catastrophic avalanches, unlike hanging glaciers, for example). The most important reason seems to be the difficulties associated with studying these features. Historical records or field-based measurements have long been the primary source of information for glacier monitoring, but IAs are often situated in cirques or niches below rock walls and steep slopes, making them relatively inaccessible [25]. Studies by [26,27] showed the use of aerial and terrestrial photographs to map glaciers and IAs, respectively. Further, the development of satellite remote sensing over the last decades has revolutionized our ability to map and monitor glaciers, as shown by the studies of [28–30]. [31,32] have also shown the potential to map and monitor small hanging glaciers and IAs from a time series of RaDAR images.

Several global products or inventories that delimit glacier outlines fairly accurately are available, such as the Global Land Ice Measurements from Space (GLIMS) [33] and the Randolf Glacier Inventory (RGI) [34]. Although the glacier outlines from global inventories are reasonably accurate for global analysis and large glaciers, their accuracy on a local scale and smaller perennial snow/ice bodies remains questionable [35], and they often consider small ice features as part of larger glacial systems [3,4]. As a result, some authors such as [36] corrected the glacier outlines (taken from the RGI) and ice divides by manual delineation, whereas others such as [37] used an automated mapping approach to correct global outlines and also generate new ice divides. The effects of significant seasonal snow and problems arising from shadows and illuminations in optical images also lead to a significant bias in glacier delineation, which can be problematic for detailed local studies [38]. The small size

of IAs thus demands high-resolution datasets for their mapping and monitoring. However, even in recent times, the dearth of freely available very-high-resolution satellite/airborne images limits the number of studies solely dedicated to small features of perennial surface ice, and IAs are under-sampled in worldwide glacier monitoring efforts [39]. In addition, the scarcity of observations from the past has led to a critical research gap in understanding the past geometry changes of small ice bodies and their mass balance.

In this context, we first aim to build a dedicated inventory of IAs for the entire MBM to redefine IAs based on their location and topographic characteristics. Since no large-scale inventories of IA are available in the literature, the study provides a first regional synthesis. Based on the IA inventory, our second goal is to understand the potential relationships of IAs with other ice masses, particularly the steep slope and cirque glaciers. Third, we assess the topographic characteristics of the IAs and discuss their topographic distribution and environmental conditions in the mountain areas.

## 2. Study Area

The MBM (Figure 1) is located in the north-western European Alps between France, Switzerland, and Italy. It is one of the Alpine external crystalline massifs situated at the footwall of the Penninic basal thrust [40]. The primary geology of the MBM is dominated by paragneisses, migmatites, and orthogneisses, with large intrusions of granitic rocks [41]. The massif is highly fractured on a large scale in multiple planes with variable density and direction. The geology and the complex topography of the terrain, combined with long periods of glacial erosion, provide relevant conditions for developing hanging glaciers and IAs on the steep rock faces.

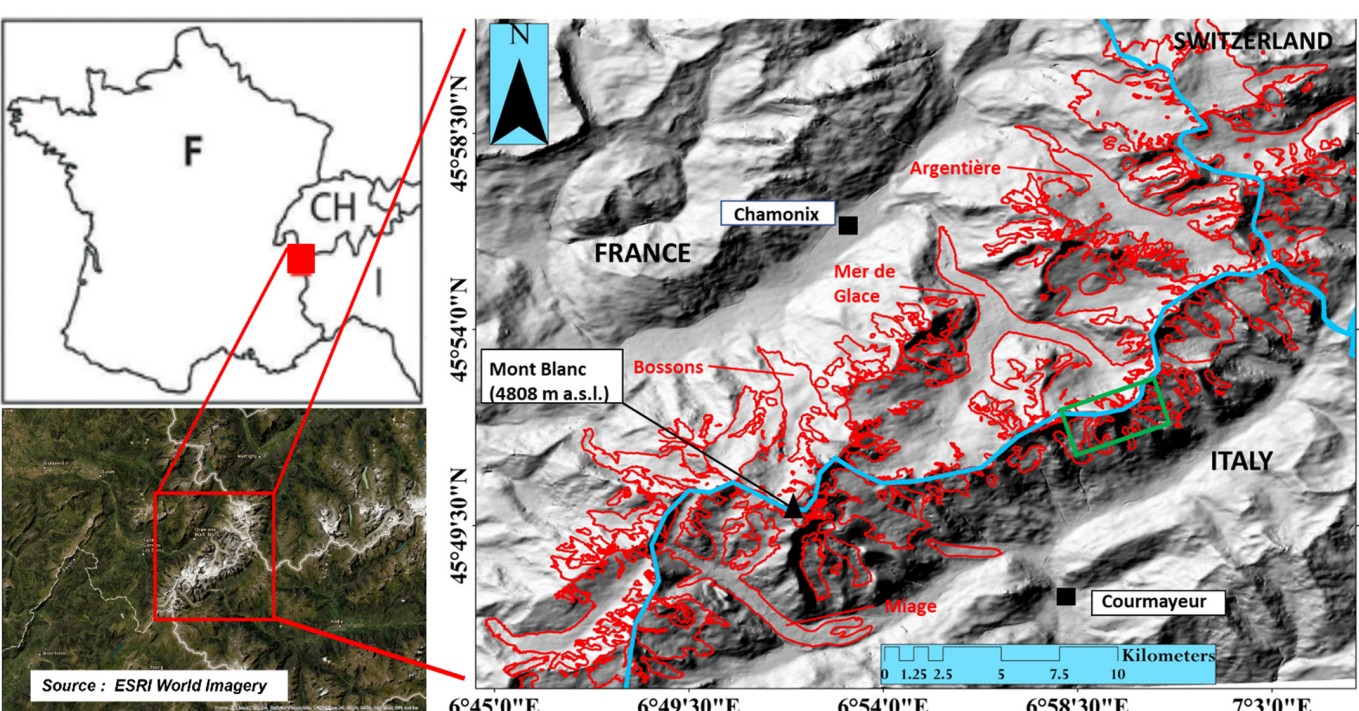

**Figure 1.** The Mont Blanc massif (MBM; Western European Alps). The hill shade is generated from the Pleiades digital elevation model (DEM), and the glacier outlines (in red) depicted are mapped from this study. The blue line shows the boundary between France, Italy, and Switzerland. The green box shows the area shown in Figure 2.

Deep valleys border the massif, which covers about 550 km². Glaciers cover about 145 km² (about 26% of the massif surface); it is the highest and most glacierized massif in the French Alps [42]. An Alpine ice cap covers Mont Blanc's summit, the highest point of the European Alps at 4808 m a.s.l. There are a dozen peaks with elevations > 4000 m a.s.l.

In addition, 12 glaciers are >5 km$^2$—bordered by steep rock walls—the largest of which is the Mer de Glace (11 km long, with an overall surface area of 30 km$^2$). The massif shows an asymmetry across the range: the Italian side is short and steep, while the French side is much longer and less steep. Thus, the largest glaciers are located on the French (NW) side, where the slopes of major valley bottoms are reduced, while glaciers are well fed by the westerly winds and melting is reduced due to the shading by the north faces. As a partial result of this asymmetry, the glacier area has decreased by 7.3% per decade on the Italian side in the last four decades, whereas it decreased by 2.3% per decade on the French side [40]. The permafrost distribution (estimated from the predicted Mean Annual Rock Surface Temperature, MARST) described by [43] shows that 45% to 79% of the total steep rock faces > 40° are underlain by permafrost. According to the aspect, the variation in permafrost distribution is evident as permafrost is mainly found at elevations above 2600 m a.s.l. in the north faces and above 3000 m in the south faces; it can be found lower in the case of favorable settings (highly fractured rock). In general, permafrost exists in all aspects and affects all rock faces at elevations above 3600 m a.s.l. [43].

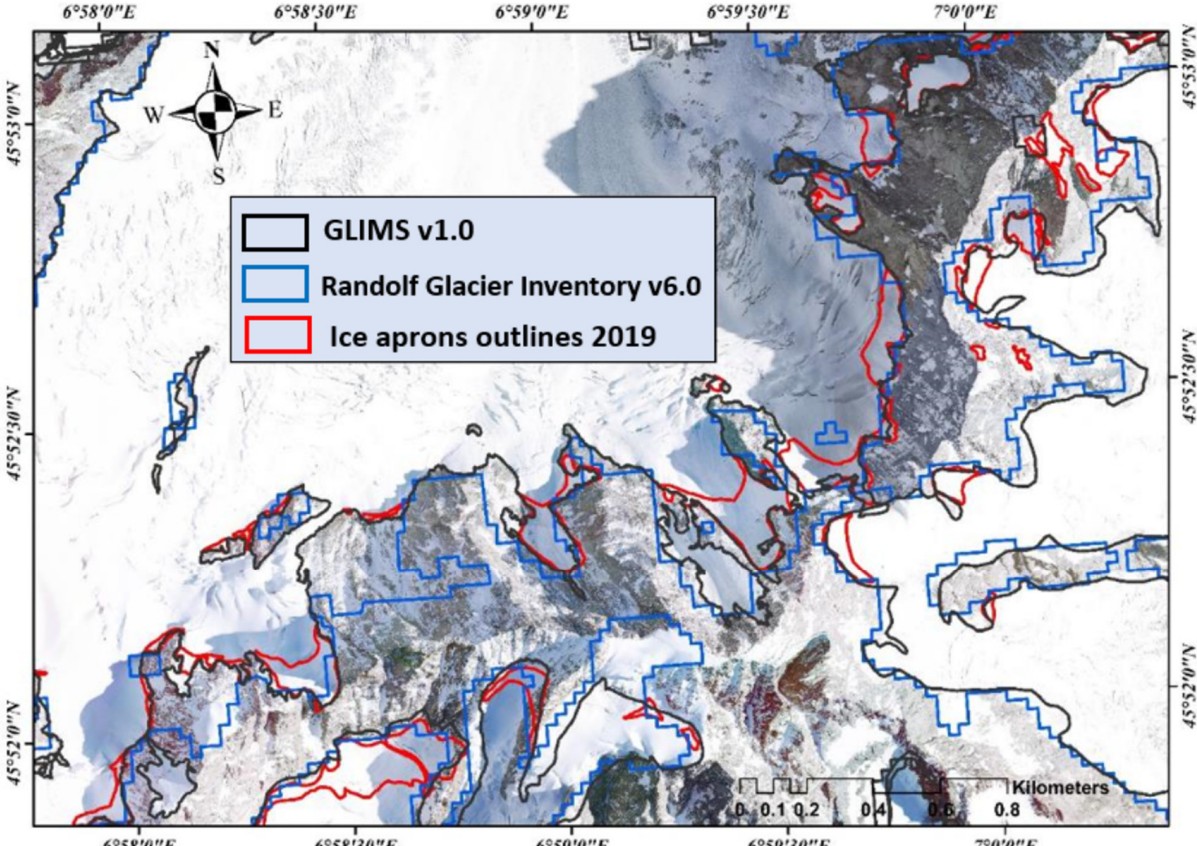

**Figure 2.** Comparison of global glacier inventories in the MBM (green box in Figure 1). The ice aprons (IAs) were delineated using high-resolution optical/aerial photographs from 2019. The RGI outlines v6.0 (blue) refer to the year 2003 [44], while the latest GLIMS outlines v1.0 (black) are for the year 2011 (submitter: Claudio Smiraglia, Università degli Studi di Milano).

## 3. Datasets

This study used optical images to build a high-resolution digital elevation model (DEM) and delineate glacier outlines to build our inventory. The optical images come from different sources, but are for the same period (except for the orthophotos from 2015). Further, existing glacier outlines such as the RGI and GLIMS (Figure 2) were used for an early reference to delineate our glacier boundaries. The various datasets used for this study are summarized in Table 1.

**Table 1.** Datasets used in the study.

| Data Type | Source | Resolution (Spatial or Temporal) | Acquisition Date |
|---|---|---|---|
| Optical | Pleiades 1A PAN | 0.5 m | 25 August 2019 |
| | Sentinel 2 | 10 m | 12 September 2019 |
| | SPOT 6 PAN | 2.2 m | 14 September 2019 |
| | Orthoimages IGN | 0.2 m | July 2015 |
| | Pleiades 1A XS | 2 m | 19 August 2012 |
| | Orthoimages IGN | 0.5 m | July 2001 |
| | Orthoimages IGN | 0.5 m | 1952 |
| Glacier inventory | Randolf Glacier Inventory (v. 6.0) | | 2016 |
| | GLIMS | | 2019 |
| Panocam (terrestrial) | Compagnie du Mont Blanc | | 2019 |

### 3.1. Optical Images

Optical data are satellite panchromatic and XS images from SPOT 6 and Pleiades at 2.2 m and 0.5 m resolution (Table 1, collected as part of the Kalideos Alps project) and IGN (French National Institute of Geographic and Forest Information) aerial orthophotos for 2015 at 0.2 m resolution. The images from different sources help reduce the reliance on a single dataset that might suffer from distortion and illumination issues. The high-resolution optical images were used to delineate the IAs and other perennial snow/ice features at a very fine scale. To validate our mapping results, we used high-resolution optical images from 1952 (orthophotos), 2001 (orthophotos), and 2012 (Pleiades PAN and XS images).

In addition, we also used terrestrial photographs from the Compagnie du Mont Blanc pano camera located at the Aiguille du Midi (3842 m a.s.l.) to show the effect of snowfall events on the overestimation of the IA area.

### 3.2. Digital Elevation Model

Considering the poor horizontal and vertical accuracies of the freely available DEMs, we built a new, very-high-resolution DEM using a pair of Pleiades stereo images. As part of the Kalideos Alps project, we acquired multi-annual stereoscopic sub-meter optical images from the Pleiades constellation over the study area. Using the rational polynomial coefficients (RPCs) provided with the Pleiades images, we computed a 4 m DEM using the Ames Stereo Pipeline (ASP) [45]. The 4 m DEM (source) was then co-registered automatically with a 2 m LiDAR DEM (reference) available for the Argentière glacier area using the methodology provided by [46]. After looking for the shift parameters, the source DEM was shifted (translation-only) using corresponding co-registration values (shift values for x, y, and z in meters). The co-registration process was stopped when an acceptable value for accuracy assessment parameters was achieved (median: −0.14 m, normalized median absolute deviation: 1.98 m). A detailed description of the DEM parameters and the accuracy assessment can be found in [47]. The high-resolution DEM was used to generate maps for topographic factors such as slope, aspect, curvature, Topographic Ruggedness Index (TRI), and permafrost distribution.

### 3.3. Global Glacier Inventories

Glacier outlines from global inventories are reasonably accurate for global analysis and large glaciers, although their accuracy on a local scale and smaller glacier bodies remain questionable [30,35]. In addition, small ice bodies such as IAs are typically under-represented in all of the global glacier datasets. Nevertheless, existing global glacier inventories provide valuable assets as an early reference to new and differentiated mapping exercises concerning various forms of perennial surface ice. Several global products or inventories that accurately delineate glacier outlines are freely available. The most

widespread ones are GLIMS and RGI, which we used as a reference to demarcate the boundaries for large glacier systems.

## 4. Description and Methods

### 4.1. Mapping of Glaciers and Snow/Ice Bodies

In order to ensure consistency and high precision throughout the mapping, the first author manually delineated all glacier outlines in a GIS environment (ArcGIS 10.6) using the optical datasets in Table 1. Since the reference datasets are of very high resolution, mapping was done at a scale of 1:1000. Although manual mapping procedures are time consuming, the operation would remain difficult with automatic mapping, especially for small snow/ice features [30,48–50]. The very high resolution of the orthophotos and optical satellite images also helps to identify and differentiate between different glaciers and other types of perennial surface ice features, identify the locations of crevasses and bergschrunds (beneficial to differentiate between glaciers and IAs), and precisely map the extent of small ice/snow features such as IAs.

Existing glacier inventories help to delineate precise past glacier outlines as they provide excellent past references [44]. The problems associated with the lack of coverage, clouds, illumination, shadow, and seasonal snow cover that make visual interpretation difficult were overcome by relying on multiple datasets instead of a single dataset. The seasonal variation of the snow cover can lead to an overestimation of the mapped snow/ice area. Hence, selecting datasets at the end of the summer period (preferably late August or early September) is crucial. The primary datasets used for this mapping exercise were Pleaides stereo and SPOT 6 images from 2019, with further validation from the orthophotographs. The orthophotos were only used for a detailed visual inspection, and no mapping exercise was performed on them, as these images come from the early part of the summer period. Considering the various optical datasets we used for our study, the images were accurately co-registered; the procedure is described in detail in [47–50]. The final glacier/ice features inventory is available for download at 10.5281/zenodo.7257980.

### 4.2. Typology of Glaciers and Perennial Surface Ice Features

The next step was to identify and classify the glaciers and other ice masses based on their characteristics. Unlike a general glacier outline, we divided the glaciers (or parts of glaciers) into various subclasses. We used a semi-automated classification method that involved masking areas based on the terrain slope, and then manually digitizing and classifying the different glacier types and other snow/ice bodies based on visual interpretations from optical images, previous literature, existing inventories, and field knowledge.

The classification of surface ice features in Alpine regions is not easy, but there is some guidance provided by the WGMS and GLIMS global datasets. Both databases provide primary classification criteria for classifying different Alpine glaciers, which proved practical for such studies. Our classification criteria are based on a combination of different parameters associated with the morphology (form, shape, size, association with other snow/ice bodies), dynamics (noticeable surface expression of movement such as crevasses in optical images), and terrain characteristics (mainly slope and location in the mountain environment). Combining the different parameters helps to distinguish ice features where only one parameter is insufficient to classify a glacier. In cases of ambiguity (i.e., for glaciers that fit into several categories), we used the suggestions provided for the primary classification of glacier types in [39]. In our classification, we prioritize the topographic characteristics, followed by the physical parameters such as the surface cover and, lastly, the general morphology of the ice feature (shape, size, and association).

#### 4.2.1. Cirque Glaciers

Cirque glaciers are found in half-open, semicircular-shaped niches or hollows located on mountainsides or in the upper part of valleys [17]. Cirques are open on the downslope side, while steep slopes usually form the other sections [6,17,51]. In our study, we classified

a glacier as a cirque glacier by observing the shape of the glacier (shaped like an amphitheater with the glacier tongue not well developed, constrained by topography at least on three sides, and a glacier width equal to or more than the length), the average slope angle (the steep side walls that constrain the glacier > 40° slope), the presence of snow/ice at the end of the ablation period, and existing literature [39,42,52].

### 4.2.2. Slope Glaciers

Regardless of their shape, size, and form, glaciers can be classified based on the underlying bedrock slope. Glaciers present on slopes with angles > 15°, referred to as steep slopes or terminating on a bedrock cliff, are called 'steep slope glaciers' [53]. They can be classified into two additional subclasses. Slope glaciers with their fronts terminating abruptly on a steep bedrock cliff are termed 'slope glaciers with a hanging front' (HF); they are commonly called 'hanging glaciers' [54] or 'avalanching glaciers' [55–57]. Sometimes, glaciers creep on steep slopes, but their fronts merge with another existing glacier body; hence, the slope glacier is not distinctly separated from other glacier systems. These are 'slope glaciers connected to other glacier systems' (CGS). A part of the Taconnaz glacier and the whole Whymper glacier are examples of HFs. Monte Bianco glacier is an example of a CGS (cirque glacier on top and slope glacier below). Differentiation from cirque glaciers is based on the overall morphology of the two glacier types and from valley glaciers by the average slope angle.

### 4.2.3. Valley Glaciers

Valley glaciers are formed when a glacier flows down a valley, erodes the side walls and the bottom, and forms a valley with steep sides and a flat base [17]. Valley glaciers can be several kilometers long, often on gentle slopes beyond the ELA. Based on the percentage of snow/ice and debris cover on the glacier surface, a further classification of valley glaciers called 'debris-covered glaciers' is possible [58,59]. However, we do not consider them as a separate class since this classification is complex and out of the scope of this paper.

We classify valley glaciers based on the average slope (<15°) [60] and the presence of snow/ice cover in the accumulation zone at the end of the summer; the glacier area being limited by topography (sidewalls) in a predefined valley; and the existing literature [39,42,44,52].

### 4.2.4. Glacierets

Glacierets are often shaped like glaciers (i.e., elongated along the slope) and have little or no movement for two consecutive years, formed because of heavy snow accumulation or left behind by receding glaciers [17]. Because of this, they are sometimes even referred to as dead glaciers (dead ice) or glacier remnants. Although glacierets resemble Alpine glaciers in shape and form, their accumulation and ablation areas are not well defined. Their thickness is also shallower than other main glaciers [61]. The classification of glacierets from other glacier types is complex, and a heavy reliance was given to on-field knowledge of the area. Our classification identified small glaciers that have ceased showing signs of displacement. The movement (or lack of any) of the glacierets was identified by observing the glacierets surface in optical images from 2019 and looking for flow indicators (such as crevasses, flow lines, and visible glacier tongue). To further validate our selection of glacierets, we verified their positions further in images from 2001 and 2011.

### 4.2.5. Alpine Ice Caps

An ice cap in Alpine environments refers to the mass of ice or snow that occupies the top of a high mountain peak [62]. They are tiny, cold ice masses that almost submerge the topography beneath [17]. In the MBM, the summit of Mont Blanc is covered with an ice cap. [63] also referred to them as 'ice fields', pointing to the summits of the Mont Blanc and Dome du Gouter. These minor ice features are periglacial, cold, and most often related to the permafrost.

Our primary basis for differentiating ice caps from other features is the morphology (dome-shaped ice masses with possible radial flow away from the centre), the presence of permanent snow/ice cover, and their locations on the summits of the highest peaks.

### 4.2.6. Ice/Snow Covers

The definition of ice/snow covers is crude and includes all parts of the steep mountain slopes covered by ice and snow. Our study defines 'ice/snow covers' as large patches of snow and ice, irregular in shape, that cannot be classified in any other class and show no movement. Ice/snow covers were mapped on images at the end of the summer to eliminate most seasonal snow in the mapped areas. Ice/snow covers were separated from IAs by looking at the steepness of the slope: they are not present on very steep slopes and are generally < 35°. Differentiation from glacierets can be tricky, but ice/snow covers generally exist on rock sidewalls where snow remains after the summer period is over, while glacierets exist in small depressions and resemble small glaciers in shape.

### 4.2.7. Ice Aprons

Finally, since the work's central theme revolves around IAs, great care was taken while identifying and accurately mapping them. As mentioned previously, no clear definition exists for defining IAs from the literature, so we consider the latest definition given by [11] as our basis for identifying and classifying IAs, which are essentially immobile small ice bodies (typically < 0.1 km$^2$) of irregular outline lying on slopes > 40°.

We selected snow/ice bodies of irregular shape that lie on mean slopes > 40° and can be mapped in continuous images from different periods. The problem of differentiating IAs from perennial snow/ice patches is critical and was mitigated using the following reasonings. We expect IAs to still exist after the summer period is over when the snow/ice patches can melt away.

This also helps to avoid overestimating the actual IA surface area (see Section 3.2); Figure 3 shows the effect of seasonal snow that can lead to a false interpretation of the actual surface area. However, this assumption is further based on the climatic conditions observed during the year of mapping. For example, snow/ice patches would continue to exist after the summer period in years when the precipitation is high and considerably cooler than decadal average summers. For this, it is decisive to check the existence of IAs in multiple images from different periods. Further, the lower limit of the IAs on the headwall of large glaciers is clear: the bergschrund is the logical limit. For our mapping exercise, the bergschrund was well visible in all of the high-resolution images since they were acquired at the end of the ablation period. In cases where this limit is not well defined or visible, we suggest considering the break in terrain slope (an area where the slope becomes < 40°) as the lower limit of the IA. For our IA inventory, we mapped ice features that could be observed in all high-resolution images from 1952, 2001, 2012, and 2019 (Figure 4). Using the methods defined above, 423 IAs were finally identified and mapped in the MBM.

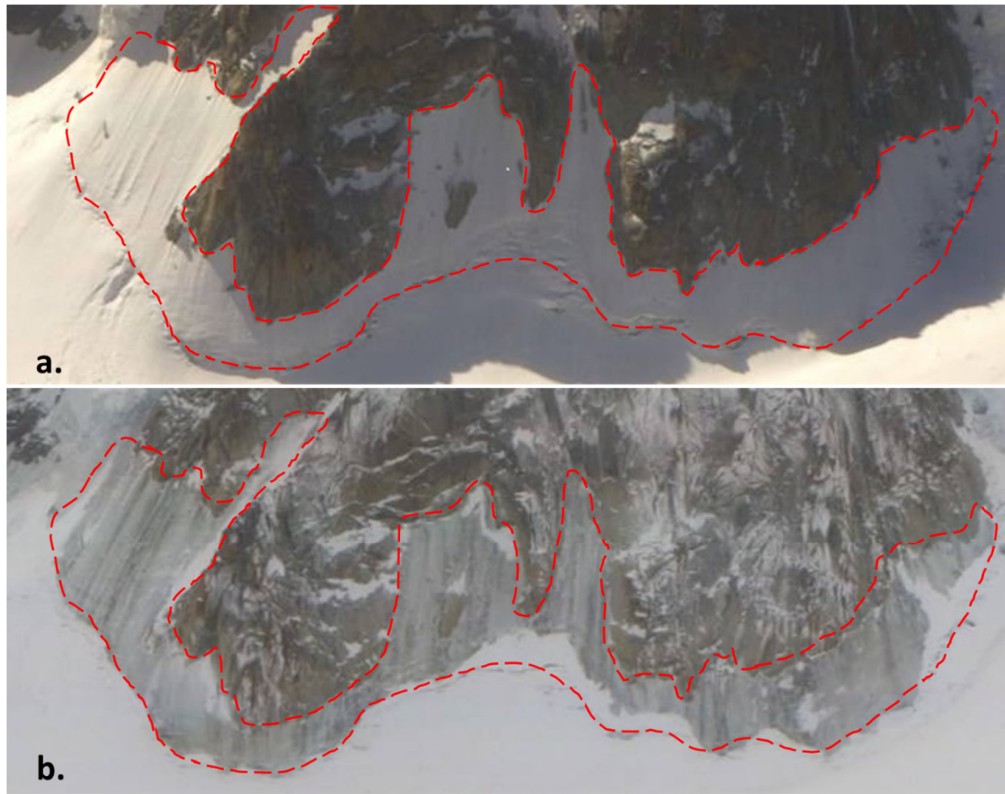

**Figure 3.** Evolution of an IA at the base of the Triangle du Tacul (3970 m a.s.l.; images are courtesy of Compagnie du Mont Blanc pano camera). (**a**) The red line is the limit of the IA at the end of August 2020 (the true extent of the IA), and (**b**) the image taken at the end of January 2021. The seasonal snowpack (white) can result in overestimating the IA area.

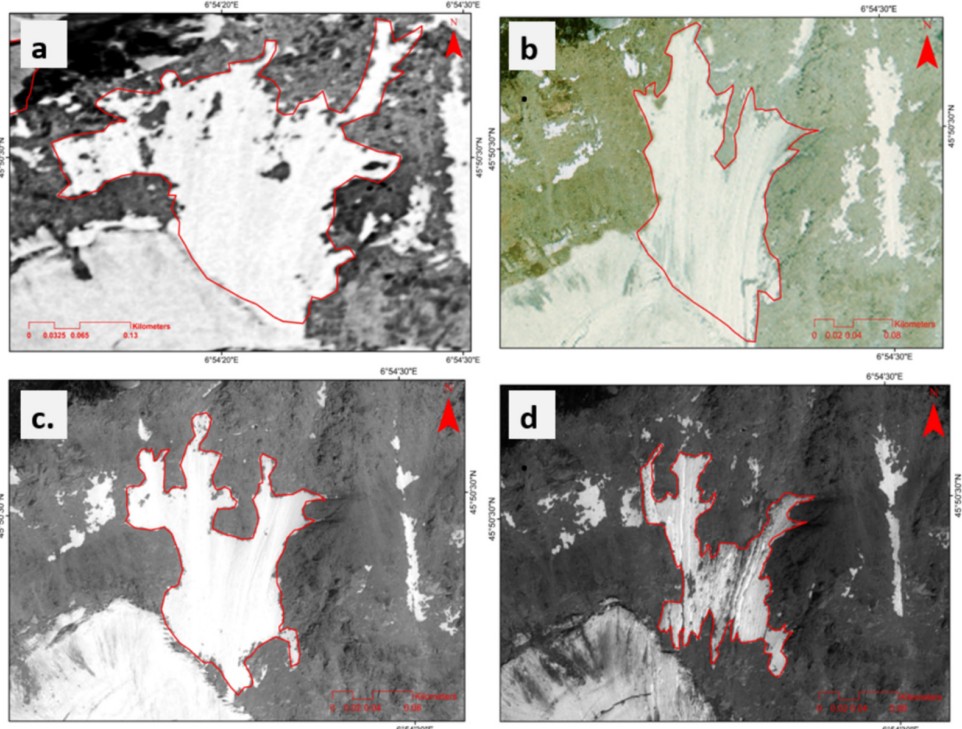

**Figure 4.** IA on the south face of the Brenva glacier on the Italian side of the massif. (**a**) IGN orthophotos

from 1952, (**b**) IGN orthophotos from 2001, (**c**) Pleiades PAN image from 2012, and (**d**) Pleiades PAN image from 2019. The red outline in each image shows the extent of the ice apron in each consecutive year, also demonstrating the area loss over the years. A quantitative assessment of the area loss of IAs over the past decade can be found in [64].

*4.3. Typology of Ice Aprons*

Based on some previous guidance by [11], IAs can be classified into three categories based on their locations in the mountain environments and their possible association with other glacier bodies.

Type 1: These IAs are formed with the decay of a larger glacier, leaving behind ice masses that show insignificant or no ice flow. These IAs emerge from the main body of the glacier as it loses mass to the point where the ice patches emerge. These IAs are thus adjacent to an existing glacier (Figure 5a).

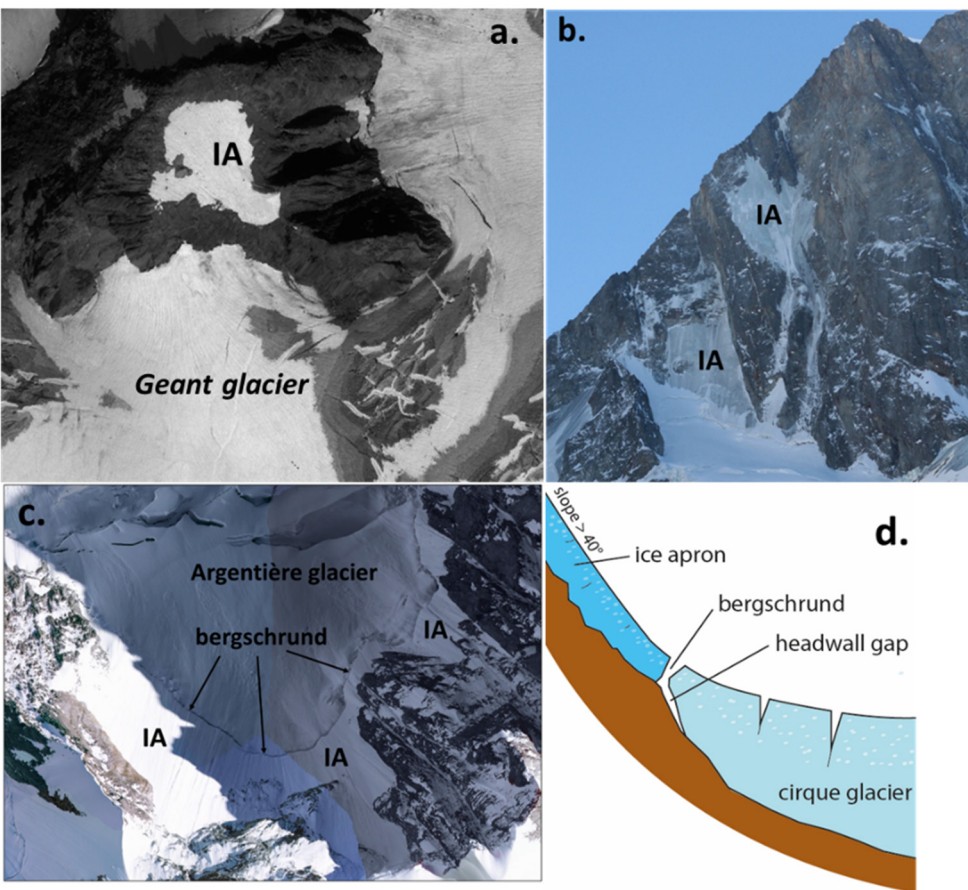

**Figure 5.** The three types of IAs. (**a**) Type 1: IA located on the south face of Gros Rognon (3541 m a.s.l.) on a rock wall associated with a large glacier; (**b**) Type 2: IA on the north face of the Grandes Jorasses (4208 m a.s.l.); and (**c**) Type 3: IAs on the headwall of the Argentière glacier (orthophotos, 2015). (**d**) An IA at the headwall of a cirque glacier.

Type 2: These IAs exist as isolated ice masses on steep rock walls. This type of IA has been generated by snowfalls that accumulate in areas where topography favors accumulation (probable concave curvature, niche, small depressions). [65] called these types of IAs 'nival ice patches' and made a strong distinction from the IAs formed in the first type based on the attributes, evolution, and genesis history of the two types (Figure 5b).

Type 3: The third type of IA exists just above a steep slope glacier (often HF) or on the top walls of a cirque glacier. These IAs are separated from a moving glacial system by the presence of a bergschrund, commonly defined as a deep crack that intersects the ice down

to the bedrock formed near the top of a glacier. It separates the moving ice (glacier below) from ice that is not moving (IA above) [66] (Figure 5c,d).

With relevance to our study, [11] recently showed that the separation between IAs and other types of glaciers can be categorical, and the bergschrund is the clear limit. They also showed that the bergschrund is relatively stable. The analysis of the ice core from the N-face of Triangle du Tacul further showed that IAs are almost immobile cold ice bodies, but present low internal deformation [22]. The IAs present on the headwall of a cirque or slope glacier thus do not participate in feeding these larger glacier systems and cannot be considered a part of the same glacier system.

Methodology for Differentiating Types of IAs

For the differentiation of IAs into different types, we used the following criteria, as also explained in Figure 6 in the flowchart:

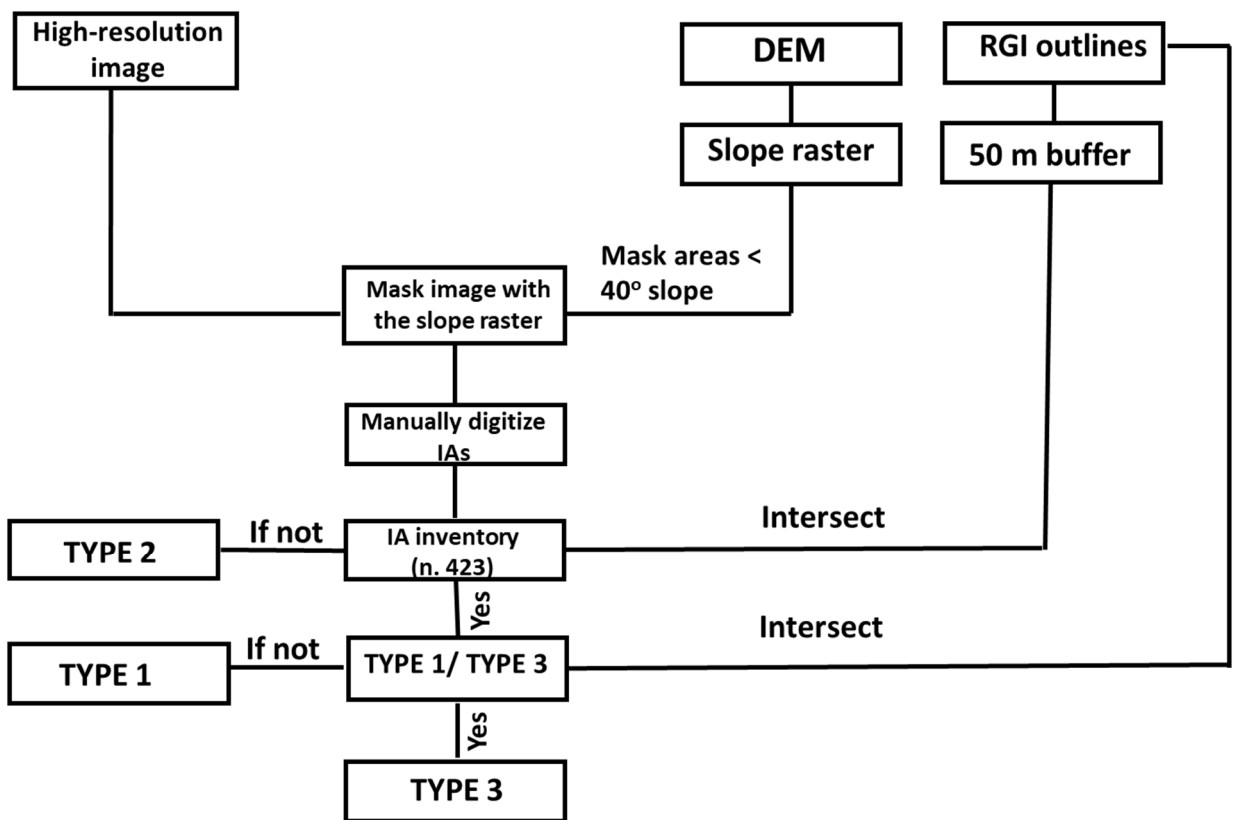

**Figure 6.** Methodology flowchart for differentiating ice aprons into different types.

1: An IA was classified into Type 1 if it fell inside the buffer of 50 m drawn around a large glacier body such as a valley, cirque, or slope glacier. We do not expect the large glacier to significantly affect smaller ice/snow bodies around them beyond this buffer. This also indicates a close relationship between the IA and a large glacier nearby.

2: An IA was classified into Type 2 if it fell outside this buffer region of 50 m drawn around the larger glacier bodies.

3: It was considered a part of Type 3 if the IA fell inside the 50 m buffer and is also a part of the predefined global glacier boundaries (RGI v6.0) since they are not considered as separate ice bodies in the RGI database. Since these IAs exist at the top of steep slope glaciers, all of these IAs are included as part of the larger glacier body outline indicated by the global inventories.

### 4.4. Generation of Topographic Parameters Used to Characterize and Map Ice Aprons

To better understand the topographic characteristics of the IAs in the MBM, we built maps for six topographic and topo-climatic factors: elevation, slope angle, aspect, curvature, Topographic Ruggedness Index (TRI), and MARST at the locations of the IAs. We normalized the area they occupy in several topo-climatic classes ($A_T$) with the total area of that class in the MBM ($A_{MBM}$) to have an unbiased assessment. A high value for this ratio (unitless) indicates that the area occupied by the IAs in that specific class is significant compared to the general area of that class present in the whole massif. All of the parameters were generated using the 4 m Pleiades DEM.

For the elevation, the DEM raster was reclassified into elevation bins of 200 m between 2600 and 4800 m a.s.l. As for the elevation, the slope raster was also reclassified into subclasses with 10° intervals. An aspect map is traditionally aggregated into eight directions, representing angular sectors of 45° with classes divided into: north (337.5°–360°, 0°–22.5°), north-east (22.5°–67.5°), east (67.5°–112.5°), south-east (112.5°–157.5°), south (157.5°–202.5°), south-west (202.5°–247.5°), west (247.5°–292.5°), and north-west (292.5°–337.5°).

Further, we generated a curvature map, estimated as a second derivative of the input surface on a cell-by-cell basis, defining the shape of the slope [67]. Generally, two types of curvatures can be determined: plan and profile. For our analysis, we only used the profile curvature as it defines the shape of the slope in the direction of the steepest slope. The profile curvature affects the flow velocity of water draining the surface; it influences erosion and deposition. In locations with convex (negative values) profile curvature, the erosion processes could prevail, while in locations with concave (positive values) curvature, the deposition process can be predominant [68,69].

TRI was calculated using the methodology proposed by [70] as a three-dimensional dispersion of vectors (x, y, and z components) normal to the grid cells considering the slope and aspect of the cell based on the spatial resolution of the DEM. The magnitude of the resultant vector in a standardized form (vector strength divided by the number of cells in the neighborhood) measures the ruggedness of the terrain. Based on the magnitude of the vectors, we divided the study region into classes of low (TRI values < 0.001), medium (0.001–0.003), high (0.003–0.005), and very high (>0.005) ruggedness. A very high ruggedness class indicates an irregular terrain variable in gradient and aspect, while a low ruggedness index class suggests regions of uniform topography. Since IA surfaces are smooth, the TRI values calculated at the IA's surface are always low. Similarly, the curvature values at the IA surface are usually close to 0, indicating a smooth surface. To avoid the discrepancy, we built a buffer of 20 m around the IA boundaries and estimated the mean TRI and curvature values from this buffer.

The MARST map was produced by implementing the 'rock surface temperature model' calibrated by [71] for the entire European Alps on the Pleiades DEM following [43]. MARST is estimated mainly as a function of two parameters: the Potential Incoming (shortwave) Solar Radiation (PISR) and the Mean Annual Air Temperature (MAAT), using a multiple linear regression model fitted with measured MARST throughout the European Alps [71]. The MARST value estimates the rock surface temperature if an ice/snow layer does not exist (the presence of ice/snow modifies the temperature fields; see [72]). Based on the MARST values, we can define the regions where permafrost is (i) cold and continuous (MARST values < −2 °C), (ii) discontinuous and close to the melting point (−2 to 0 °C), (iii) only sporadic (0 to 3 °C), and (iv) not expected (>3 °C) [71,73].

### 4.5. Uncertainty Associated with Mapping Ice Aprons

The subjective interpretation and generalization result in uncertainties during the manual delineation of the outlines [44]. Estimating the uncertainties in mapping estimates is tricky; however, some guidelines for the quality assessment of manual glacier area determination are provided by [50]. Considering this, the first author performed manual digitization three times for 150 IAs ranging from 0.0001 $km^2$ to 0.01 $km^2$. The IAs were selected according to different challenges associated with manual digitization, mainly in

shadow and highly complex topography areas. An uncertainty estimate is given through the Mean Area Deviation (MAD) in percentage considering three digitizations, taking the first digitization as a reference. The MAD provides a percentage estimate of how the final area calculated varies across multiple digitizations for each polygon. The MAD values are affected by the size of the polygon manually digitized. Previously, authors such as [3,12,50] reported an increase in the uncertainty of manual digitizations with a decrease in the size of the polygons. To test the same, we divided our sample size into sub-classes based on the IA's surface area < 0.003 km$^2$, 0.003–0.007 km$^2$, and >0.007 km$^2$ and the MAD values for each sub-class of IAs based on their area were estimated.

## 5. Results

### *5.1. High-Resolution Glacier Typology Map of the Mont Blanc Massif*

Figure 7 shows the typology map of the MBM glaciers, and Table 2 displays the total area occupied by each sub-class. The total area occupied by all glaciers/ice bodies is 161 km$^2$, which is ~30% of the total area of the massif. We recognized nine valley glaciers, the most dominant type in terms of area, occupying more than half the total area occupied by the glaciers/ice bodies (~52%). Slope glaciers are the second most dominant category (~24% of the total area), while cirque glaciers are third (~11%). The median size of valley glaciers is also the largest (~7.75 km$^2$), while that of cirque glaciers is the second largest (~1.04 km$^2$). The Mer de Glace is the largest valley glacier in the MBM, with a total surface area of ~29.2 km$^2$. For glaciers, in terms of numbers, the slope glaciers are the most dominant (*n.* 102). Out of the 102 slope glaciers, 45 are hanging glaciers, 12 are connected to other glacial systems, and the remaining 45 are typical slope glaciers that cannot be subdivided into additional classes. Regarding the average area of each slope glacier type, hanging glaciers are the smallest (~0.04 km$^2$ median area), while the typical slope glaciers are the largest (~0.41 km$^2$). The Taconnaz glacier on the French side of the massif is the largest slope glacier (4.49 km$^2$). We identified six ice caps, totaling an area of 0.44 km$^2$, with the largest being located at the summit of La Grande Bosse on the NW Mont Blanc ridge, occupying an area of 0.262 km$^2$, followed by the ice cap of the Mont Blanc with an area of 0.239 km$^2$. We also mapped 43 glacierets with a total area of 3.49 km$^2$. The glacieret in front of the Grands glacier is the largest, with an area of 0.40 km$^2$, almost five times the median size (~0.053 km$^2$) of other glacierets. Moreover, 109 snow/ice covers were mapped with a total area of 6.18 km$^2$ (~3.83% of the total glacier/ice bodies area). The median size of individual snow/ice cover is 0.029 km$^2$, almost ten times larger than the median size of IAs. This is also a significant factor for differentiating snow/ice covers from IAs, as they are much larger than IAs. Finally, 423 IAs were identified in the MBM, occupying 4.214 km$^2$ in 2019, which is ~2.5% of the total glacier/ice bodies area.

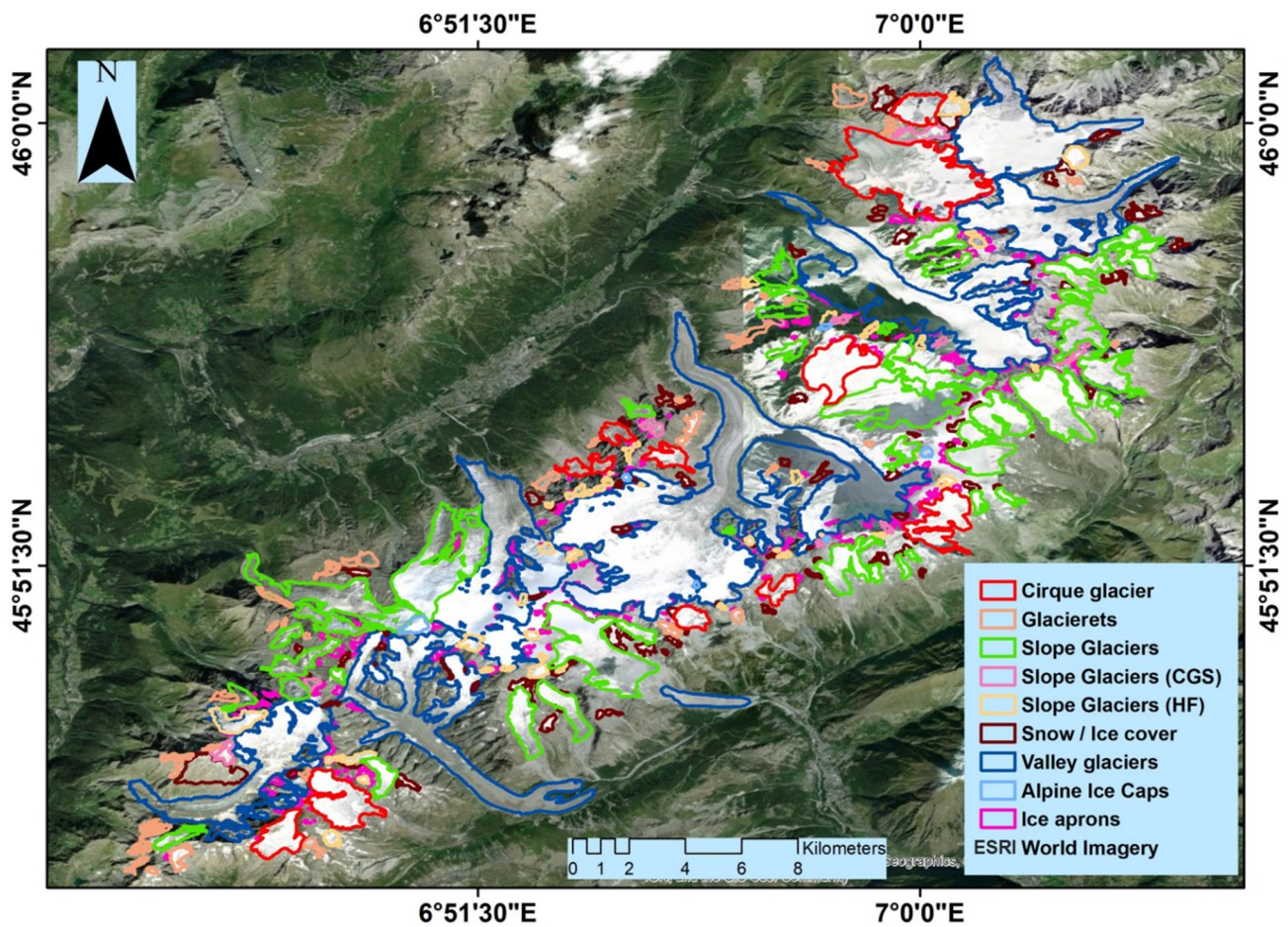

**Figure 7.** High-resolution typology map of the glaciers of the MBM.

**Table 2.** Glacier types, numbers, and areas in the MBM.

| Types of Glaciers | Number in the MBM | Total Area (km²) | Percentage of the Total Glacier Surface Area | Minimum Area (km²) | Maximum Area (km²) | Average Area (km²) | Median (km²) |
|---|---|---|---|---|---|---|---|
| Cirque glaciers | 11 | 18.10 | 11.18 | 0.39 | 6.91 | 1.63 | 1.04 |
| Slope glaciers | 45 | 39.4 | 24.47 | 0.033 | 4.49 | 0.87 | 0.41 |
| Slope glaciers (HF) | 45 | 3.85 | 2.39 | 0.003 | 0.65 | 0.08 | 0.041 |
| Slope glaciers (CGS) | 12 | 1.45 | 0.90 | 0.02 | 0.46 | 0.12 | 0.08 |
| Valley glaciers | 9 | 85.3 | 52.3 | 0.78 | 29.2 | 9.42 | 7.75 |
| Glacierets | 43 | 3.49 | 2.15 | 0.007 | 0.40 | 0.08 | 0.053 |
| Ice aprons | 423 | 4.21 | 2.57 | 0.0001 | 0.10 | 0.009 | 0.003 |
| Snow/ice covers | 109 | 6.18 | 3.83 | 0.0006 | 0.299 | 0.05 | 0.029 |
| Ice caps | 6 | 0.44 | 0.27 | 0.017 | 0.262 | 0.07 | 0.034 |

The observation from the uncertainty analysis (Figure 8) shows that for our database of IAs, the MAD values increase with the decrease in the IA area. Overall, the mean MAD for all samples and multiple digitizations was ±6.3%. For samples in sub-class < 0.003 km², the MAD is ±7.2%, while for sub-class 0.003–0.007 km², it is ±6.4%. The MAD value drops further to ±5.2% for area class > 0.007 km².

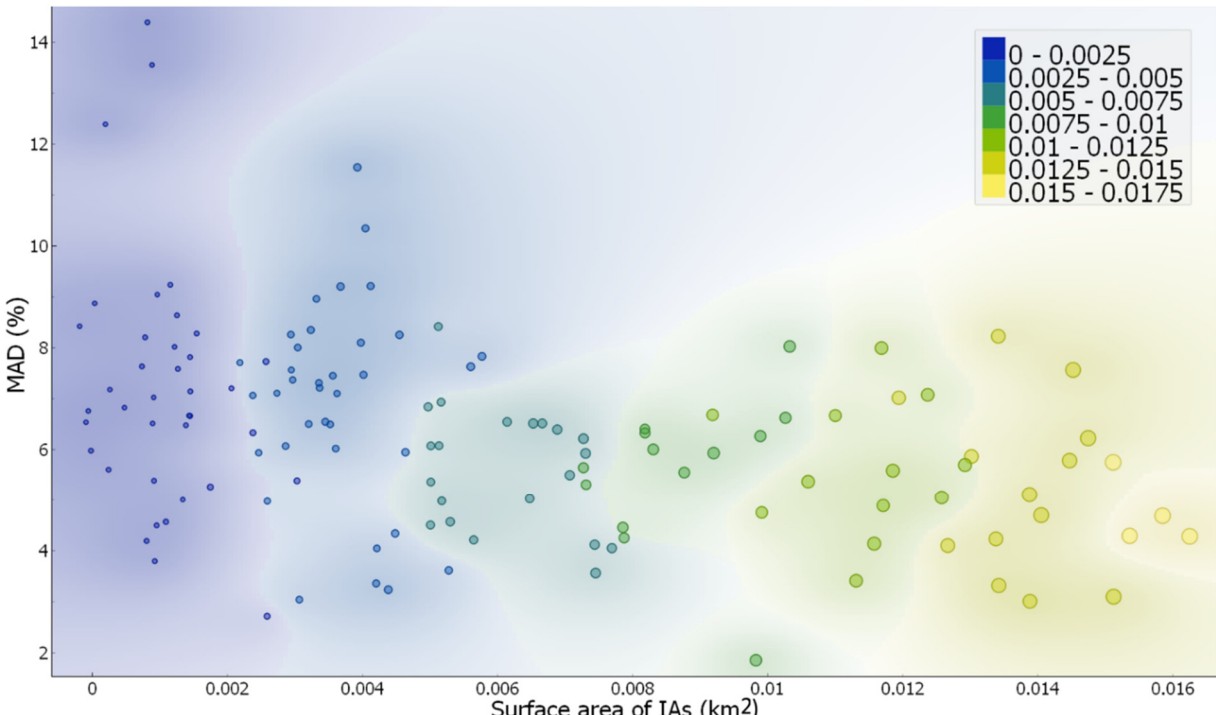

**Figure 8.** The plot's y-axis shows the mean area deviation (MAD) values in % for 150 multiple-digitized IAs. The x-axis is the surface area of IAs in km². The color regions inside the plot and the tick mark diameter both depict the size class of IAs, as represented in the label. The darker colours (blue to green) suggest the smaller size of IAs, while the lighter colours (yellow) show large IAs.

*5.2. Location and Morphometric Characteristics of Ice Aprons*

The analysis of IAs with elevation shows that the belt 4100–4300 m a.s.l. is dominant (Figure 9a), followed by 4300–4500 and 3300–3900 m. The lowest elevation class (2700–2900 m) is the least represented category. The comparison presents a trend of gradual increase with elevation, with exceptions at 3900–4100 and 4300–4500 m a.s.l. In general, IAs dominantly exist at high elevations (median elevation = 3375 m a.s.l., Table 3) above the regional ELA (~3200 m a.s.l.). A comparison with the slope angles shows that the 60–70° class is the most important, closely followed by 50–60° and 70–80° (Figure 9b). The class 40–50° is the least represented. Thus, IAs preferably exist on steep slopes with median slope values of the dataset at 61° (Table 3).

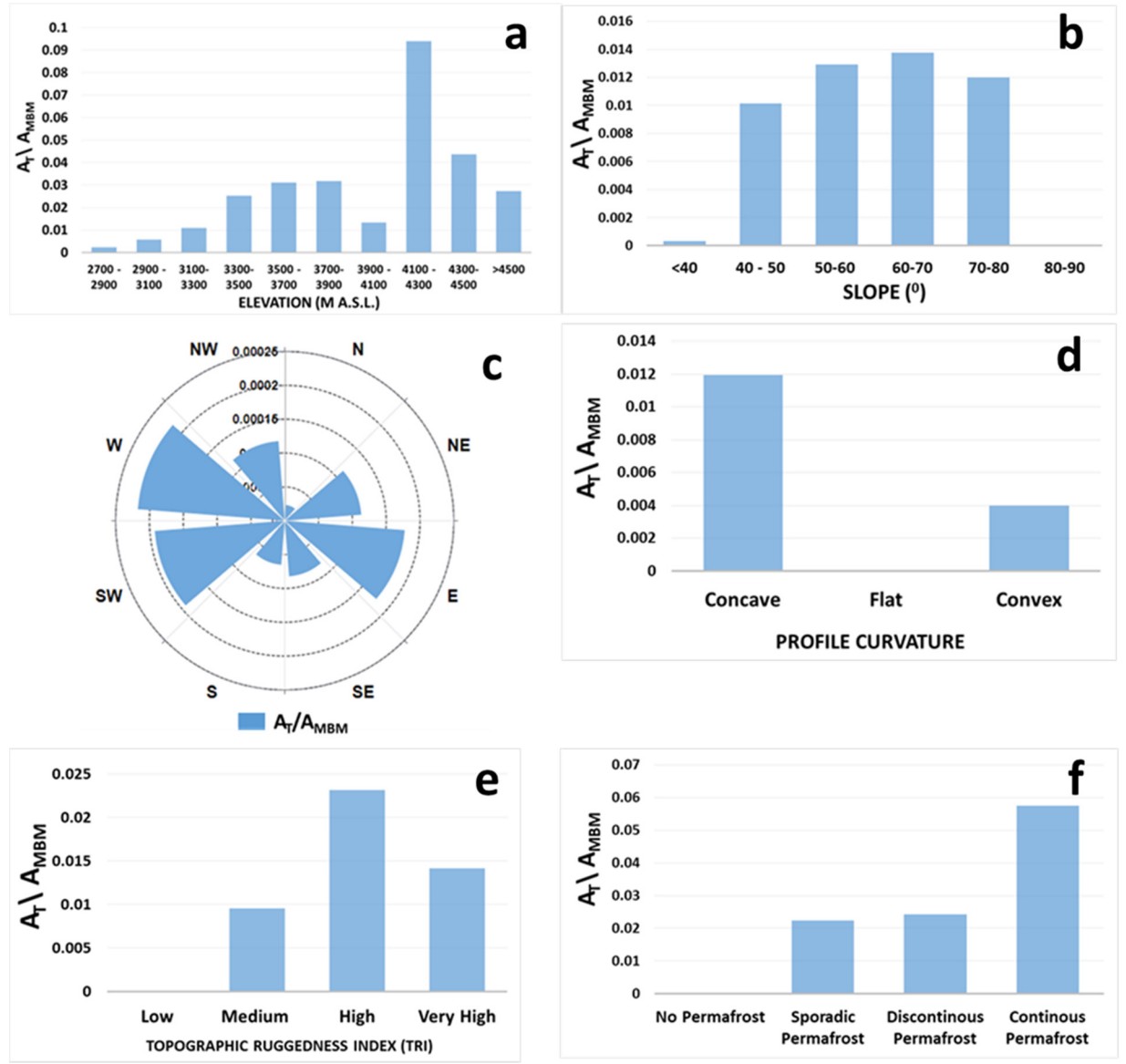

**Figure 9.** Morphometric characteristics of the IAs in the MBM based on different topographic and topo-climatic parameters: (**a**). elevation, (**b**). slope, (**c**). aspect, (**d**). curvature, (**e**). TRI and (**f**). permafrost. $A_T$ refers to the area of the IAs in a particular topo-climatic class, and $A_{MBM}$ is the total area of that class in the MBM.

**Table 3.** Statistical analysis of the IA dataset with different topographic parameters.

|  | Elevation (m a.s.l.) | Slope (°) | Aspect | Curvature | TRI | MARST (°C) |
|---|---|---|---|---|---|---|
| Min. value | 2721 | 39 | 0.35 | −4.665 | 0.001 | −9.84 |
| Max. value | 4590 | 78 | 359 | 5.626 | 0.007 | 4.52 |
| Average | 3398 | 58 | 312 | 0.086 | 0.004 | −2.1 |
| Median | 3375 | 61 | 310 | 0.042 | 0.004 | −2.14 |

Additionally, 192 IAs (~46% of the total number) with a total area of 1.31 km$^2$ (31% of the total IA area) are found in the northern aspects (NE, NW, and N), while the total area of these aspects in the MBM is 244 km$^2$ (~44.5% of the total area of MBM). After normalization, the best-represented aspects in terms of surface area are W, SW, and E

(Figure 9c), while IAs on the shaded northern aspects (NW, NE, and N) are less present. The concave profile curvature class is more significant than the convex and smooth ones (Figure 9d). The median curvature value observed for the dataset is 0.042 (Table 3), which suggests mainly concave curvatures. Subsequent comparison with the TRI shows that the IAs exist in high- and very high-ruggedness topographies (median = 0.004, Table 3). Low-ruggedness topographies are the least favorable locations for the existing IAs (Figure 9e). Furthermore, a final comparison with MARST shows a close relation between IAs and the 'continuous permafrost' as this class is most dominant (median MARST values = −2.14 °C, Table 3), followed by the 'discontinuous' and 'sporadic permafrost' classes (Figure 9f). The 'no permafrost' class is insignificant, very likely indicating that IAs only exist with underlying permafrost.

### 5.3. Ice Apron Typology and Their Distribution in the MBM

Based on the types of IAs described in Section 4.3, 63 IAs (~15%) can be classified as Type 1 (exist in proximity to a large glacier) IAs, while only ~3% (12 out of 423) can be classified as Type 2 (isolated IAs). However, regarding the percentage of the total IA area, Type 1 IAs occupy ~13.3%, while Type 2 IAs occupy just 2% of the total IA area. Out of 423 IAs, 348 (~82%) fall into the Type 3 (present on the headwalls of glaciers, separated by a bergschrund) category, occupying around 84.5% of the total IA area, making it the most dominant type of IA in the MBM (Table 4). The median size of Type 3 IAs is also the highest (~0.006 km²), while that of the Type 2 IAs (IAs on isolated steep slopes) is the lowest. Further, Type 3 IAs are also located at the highest median elevation (~3452 m a.s.l.); they are also present on the steepest slopes (~63° median slope) and present the lowest MARST values (~−2.63 °C) (Table 5). Type 2 IAs, on the other hand, show the highest TRI values, indicating that they are present on isolated rock faces surrounded by the most rugged terrains. Type 2 IAs also show the highest MARST values (~−1.52 °C) (Table 5), making them the most vulnerable to permafrost degradation effects.

**Table 4.** Type-wise distribution of IAs in the MBM.

| Type of IA | Number in the MBM | Total Area (km²) | Percentage of the Total IA Area | Minimum Area (km²) | Maximum Area (km²) | Average Area (km²) | Median (km²) |
|---|---|---|---|---|---|---|---|
| Type 1 | 63 | 0.56 | 13.30 | 0.001 | 0.015 | 0.0085 | 0.004 |
| Type 2 | 12 | 0.084 | 2.00 | 0.0001 | 0.012 | 0.0072 | 0.003 |
| Type 3 | 348 | 3.56 | 84.56 | 0.002 | 0.10 | 0.0102 | 0.006 |

**Table 5.** Statistical analysis of the IA types with different topographic parameters.

| Type of IA | Median Elevation (m a.s.l.) | Median Slope (°) | Median Aspect | Median Curvature | Median TRI | Median MARST (°C) |
|---|---|---|---|---|---|---|
| Type 1 | 3210 | 58 | 301 | −0.025 | 0.0041 | −1.89 |
| Type 2 | 3381 | 60 | 285 | 0.053 | 0.0055 | −1.52 |
| Type 3 | 3452 | 63 | 335 | 0.048 | 0.0043 | −2.63 |

### 5.4. Size of Ice Aprons

Most IAs are extremely small in area, as most (~43%) fall in the size class < 0.003 km², while more than 70% of the IAs are less than 0.009 km² (Figure 10a). The mean size of IAs shows an almost increasing size trend with the elevation (except for the highest elevation class). The largest IAs are found in the elevation belt of 3700–3900 m a.s.l. The smallest IAs are found at the least elevation class (2700–2900 m) (Figure 10b). A similar comparison

with the slope (Figure 10c) shows a near-inverse relation of slope angle with the size of the IAs: as the slope steepness increases, the average size of the IAs decreases. The largest IAs are found in the lowest slope angle class, while the smallest IAs show the steepest slope angles. This trend is almost continuous except for the slope class 40–50°, where the mean size of IAs is lower than that in the next class. However, this observed variation is insignificant, as we do not observe a large difference in IA sizes and increasing slope angles. A comparison of the IA size with the aspect shows that the largest IAs are found on the SE aspect, followed by the N and the E aspects (Figure 10d). The smallest IAs are found in the NW aspect.

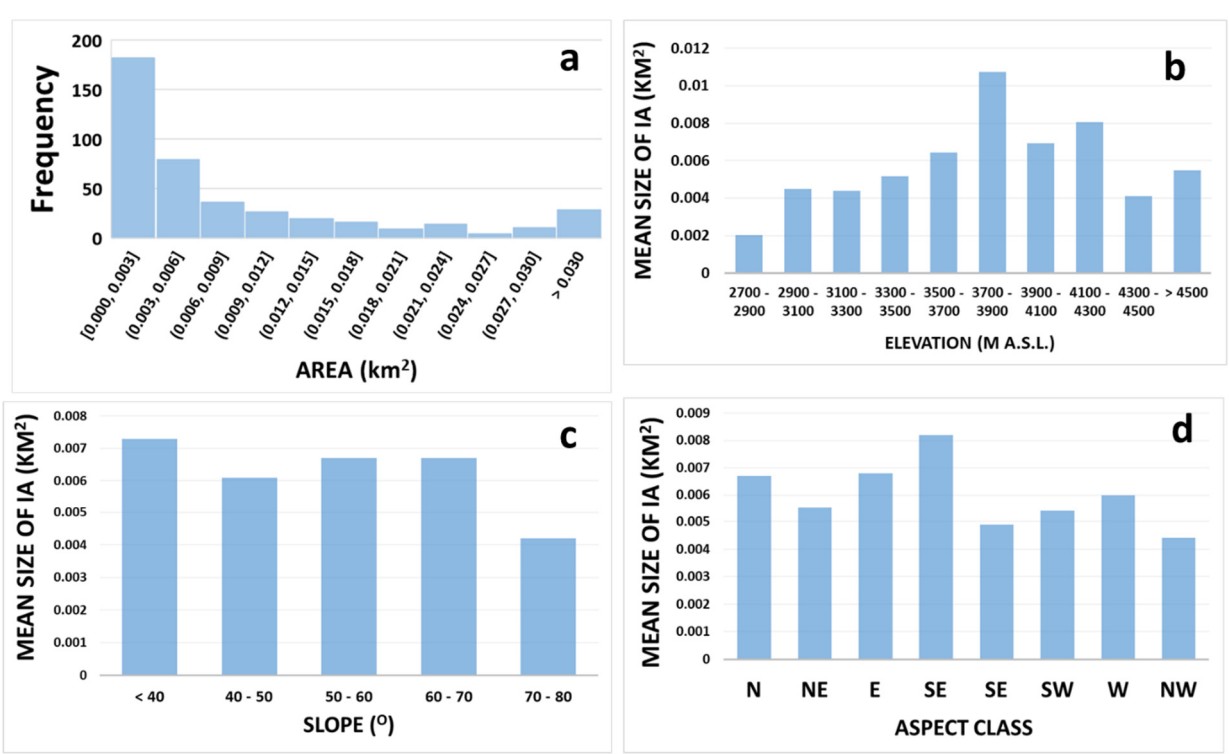

**Figure 10.** (**a**) Frequency distribution of the IAs; comparison of the IAs mean area with (**b**) elevation, (**c**) aspect, and (**d**) slope angle.

## 6. Discussion

This first analysis of the glacier/ice bodies inventory shows that IAs are a dominant feature in the MBM according to the absolute numbers. The total number of IAs mapped in the MBM is significantly higher than any other glacier or ice/snow feature (Table 2). This proves an aforementioned point: small ice bodies (such as IAs) account for a majority (~80–90%) of the total number of mapped ice features in the MBM. This result is consistent with the findings of [1]. However, the absolute number of IAs does not reflect the percentage of the total glacier area they occupy in the MBM. IAs occupy just a tiny fraction of the total glacier/ice body area (2.57% of the total glacier area in the MBM), showing their size is very small compared to other features (0.003 km$^2$ median size compared to 7.75 km$^2$ for valley glaciers).

IAs on the headwall of steep slope glaciers (Type 3) are the most dominant type of IA in the MBM. Throughout the article, it was emphasized that IAs are not glaciers, nor are they physically connected to them. Therefore, they belong not to the glacial, but the periglacial realm. The periglacial zone is itself divided into the 'upper periglacial belt' covering the highest ridges and peaks above the ELA of the glacier, and the much better understood 'lower periglacial belt' below the ELA [74,75]. IAs are part of the upper periglacial belt. This further makes a case for improving our understanding of these small ice bodies and distinguishing them as separate entities in global inventories. In addition, Type 2 IAs are

the rarest in the MBM and have the least average size of all categories of IAs. These isolated IAs are probably the most vulnerable to global warming. As a result, these IAs are generally found in topographies that could favor snow accumulation, such as in small depressions and where ablation rates are low (mainly well above the regional ELA line).

A few observations are evident from comparing the IAs with the different topographic factors. A comparison with elevation shows that when placing the regional ELA conservatively at ~3200 m a.s.l. [74], we can state that IAs predominantly exist at elevations above the ELA, confirming the definition given by [11,16]. This is significant, since elevation strongly influences climatic conditions (temperature, precipitation, and wind speeds) and permafrost; this likely strongly influences the durability of IAs in the context of changing climate. Previous research has also shown that more convex curvatures are observed at higher elevations, while concave curvatures are predominant at lower elevations [76]. Concave curvatures generally favor snow depositional processes; hence, IAs are more predominant in the middle-high altitudes than in the highest elevation classes. Additional analysis showed a general increase in the IAs size with an increase in elevation, but the largest IAs are found in the middle-high elevations (3700–3900 m a.s.l). However, high elevations also experience more pronounced wind speeds [77], resulting in the wind transportation of snow. This could make it harder for the snow to accumulate on steep slopes, as the high-speed blowing winds tend to transport snow from higher elevations and deposit them on gentler slopes at lower elevations, as shown by [78,79]. This may partly explain why we do not see the largest IAs in the highest elevation class. It can be stressed that the middle-high elevations may be the most conducive to the formation/durability of large IAs.

A comparison with slope angles shows that IAs exist on steep slopes (40 to 65°) at high altitudes. The steepest slopes (close to vertical) do not favor long-lasting snow accumulation because the amount of snow accumulation is inversely proportional to the slope angle, and above a critical threshold, no snow accumulates permanently [80]. The comparison of slope angles with elevation showed that terrain slope steepness is directly proportional to elevation, i.e., higher elevation terrains are steeper. However, bedrock slope is inversely proportional to the size of the IAs, as smaller IAs are present on very steep slopes. This can be explained because as slope steepness increases, slopes become more avalanche-prone, resulting in mass wasting. Previous studies have shown that the glacier surface area loss increases with slope steepness [81–83]. This is probably why the thickness of IAs from previous studies is reported to be significantly lower than other glacier/ice bodies in mountain environments (e.g., [22]).

Further, only snowfall occurring at temperatures between −5 and 0 °C is believed to accumulate on very steep slopes if we exclude wind drift effects [84,85]. As a result, only limited snow can accumulate on these steep slopes, while ablation effects continue to increase (with a rise in temperatures). This can lead to a situation where the ablation rates for IAs exceed the accumulation rates, leaving more IAs vulnerable to climate change impacts in the future [11].

In addition, the most dominant aspect classes are W, SW, and E. It is noteworthy that the west-facing slopes receive the most direct solar radiation during the hottest time of the day in the northern hemisphere [86,87]. Moreover, the south-facing slopes generally receive more solar radiation than others for a given surface. From further analysis of the aspect, we observed that IAs on N-facing aspects exist at lower altitudes than those on S: IAs on the S aspects can only survive at very high altitudes. As stated before, the asymmetry of the MBM is evident with S slopes more rugged and steep than the N slopes. E-facing IAs also show lower MARST values than the W-facing IAs, further corroborating the results of [43,87].

Interestingly, S-facing IAs show more convex terrain curvatures than the N-aspect IAs (concave curvatures). As a result, except for the SE-facing IAs, we noticed that the S-aspect IAs are generally smaller than the N-facing IAs.

A comparison with mean curvature classes shows that most IAs exist in areas surrounded by concave terrains. This is part of the reason for the formation of the IAs: concave profile curvatures favor snow accumulation over long periods. Thus, snow can accumulate, and the melting and refreezing of the water-saturated snow eventually leads to the formation of a stable ice mass. IAs also exist preferably on rugged terrains, as compared to smooth terrains. High-roughness terrains may first retain the snowpack, favoring the snow metamorphism processes [88] and then blocking the downward movement of the ice [89,90]. In addition, the low thickness of the snowpack also prevents creep events. However, over time, the balancing effects vanish if the snowpack is sufficiently large to form a smooth surface at the base of the bedrock, which can trigger an avalanche.

The last analysis with MARST shows that IAs exist on permafrost. Permafrost processes and their relationship with surface perennial ice features is an essential research topic in the permafrost community. Perennial surface ice features, such as IAs, have been utilized as a potentially strong indicator of the presence of permafrost [91]. IAs are cold and frozen ice masses that stick to the subsurface because their ice temperatures cannot exceed 0° C in summer, while they cool far below 0° C in the winter. The resulting frozen conditions stabilize ice on steep slopes and help the IAs sustain in such topographies without sliding. This phenomenon was further evidenced [22], who reported temperatures < 0 °C at the base of the ice core taken in the Triangle du Tacul. The frozen conditions prevent subsurface rock destruction through freeze–thaw processes. With the disappearance of the IAs, the daily and seasonal freeze–thaw processes would initiate the formation of the active layer, resulting in increasing rock slope instabilities [87]. Over extended timescales (decades to centuries), permafrost originally covered by cold IAs will likely warm up to greater depths, progressively destabilizing larger masses of over-steepened bedrock. This is already experienced by the increasing number of rockfall events in the study region, as noted by [20].

## 7. Conclusions

We build a new inventory for the MBM using a combination of high-resolution satellite and aerial images. It encompasses different glacier and snow/ice bodies to understand the relations of the IAs with other perennial surface ice features and provides a detailed characterization of their morphology and topo-climatic settings. The final glacier/ice body inventory is the only one available at such a high resolution with a sub-classification of various perennial surface ice features at a regional scale. Compared to all mapped features, the valley glaciers are the most dominant category (in terms of the percentage of the total perennial ice surface area) in the MBM, occupying ~52% of the total glacier/ice body area of the massif. This is followed by the slope glaciers and the cirque glaciers, which occupy ~28 and ~11% of the total glacier area. Other features such as slope glaciers with hanging fronts, slope glaciers connected to other glacier systems, glacierets, snow/ice covers, and Alpine ice caps were also identified and delineated. We mapped 423 IAs in the MBM. The total area occupied by the IAs is 4.21 km$^2$, which is only 2.57% of the total glacier area of the massif. Most of the IAs are small (<0.009 km$^2$). As a result, their mapping and monitoring can be highly challenging. However, this study shows that this can be achieved with continuous high-resolution satellite and aerial images. The IAs inventory provided the opportunity to understand some topo-climatic factors explaining their location in high Alpine environments. This also gave us a few insights into the evolution and formation of the IAs. Utilizing our extensive database, we can elaborate on the existing definition of IAs given by [11] with proof from a regional analysis. We thus redefine IAs as 'extremely small' (generally < 0.009 km$^2$) and thin ice patches of irregular shape, existing above the regional ELA, on very steep slopes (generally between 40 and 65°), surrounded by concave curvatures and rugged terrains in the permafrost areas. A comparison of the absolute size of the IAs with the most critical topographic parameters (elevation, aspect, and slope) shows that large IAs exist at mid-high elevations (3700–3900 m a.s.l.), on slopes between 50 and 60°, on SE, N, and E facing slopes.

Most of the IAs exist in relation to large glaciers, as ~82% of all IAs are located on the headwalls of the valley, slope, or cirque glaciers. They are also the largest type of IAs in the MBM (median size ~ 0.006 km$^2$). These IAs are separated from the main glacier body by a bergschrund. As all global glacier inventories do not consider IAs a separate entity or show them as part of the large glacial systems, there is a need to redefine glacier inventories by considering IAs as distinct ice features. This is part of the reason why IAs have been ignored by glaciologists even as they stand out as a critical component of the high Alpine cryosphere system. The small size and thickness of IAs, coupled with the global warming context, imply that most IAs are facing a risk of complete disappearance in the coming decades. This shrinkage/surface area loss of IAs also leads to increasing difficulties for mountaineering practices and rockfall risks. Further, the disappearance of these ice masses represents the loss of a crucial glacial heritage as they preserve ice that is several hundred to several thousand years old. Hopefully, this first detailed research on the IAs at a regional scale will lead to improved awareness of the scientific community to focus their attention on these small, but critical, glacial features.

**Author Contributions:** S.K. and L.R. designed the study. S.K. drafted the paper, which all co-authors revised. L.R. and F.M. helped in data interpretation and analysis. Y.Y. and E.T. proofread the manuscript and provided valuable input for improving the overall quality of the paper. All authors have read and agreed to the published version of the manuscript.

**Funding:** This research was funded by ANR, grant number 19-CE01-18 and the APC was funded by the project ANR 19-CE01-18 Wisper.

**Data Availability Statement:** The detailed glacier inventory can be found for free download at https://doi.org/10.5281/zenodo.7257980 (accessed on 27 October 2022).

**Acknowledgments:** This research is part of the USMB Couv2Glas and GPClim projects. Pleiades data were acquired as part of the Kalideos-Alpes project and successfully processed under the "Emerging risks related to the 'dark side' of the Alpine cryosphere" program. We also thank C. Vincent of the *Institute of Environmental Geosciences* (IGE, Grenoble, France) for providing the Argentière glacier DEM.

**Conflicts of Interest:** The authors declare that they have no conflict of interest.

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
