# Peer review of "Ice Aprons in the Mont Blanc Massif (Western European Alps): Topographic Characteristics and Relations with Glaciers and Other Types of Perennial Surface Ice Features"

_remotesensing, doi:10.3390/rs14215557_

Round 1

Reviewer 1 Report

Comments by Wilfried Haeberli

on

Ice aprons in the Mont Blanc massif (western European Alps): Topographic characteristics and relations with other types of glaciers

Paper submitted to remote sensing by

S. Kaushik, L. Ravanel, F. Magnin, E. Trouve and Y. Yan

General

The submitted paper maps and statistically analyses in a detailed and differentiated way various features of surface ice based on high-resolution optical satellite images. The focus of the investigation is on ice aprons (IAs) in steepest terrain, a clearly under-researched phenomenon of cold mountains on Earth. In this sense, the study is most welcome and can help setting the stage for more widespread and more appropriate treatment of this climate-sensitive aspect of surface ice in rugged mountain topography. The material is well presented and the text is clearly written following a logical structure. Essential improvements and more critical reflections are primarily necessary concerning (a) the terminology/semantics used with respect to the terms “glacier” and “glacial”, and (b) a more appropriate treatment of permafrost, the basic physics involved and periglacial conditions.

(a)   Terminology and semantics

The authors use the terms “glacier” and “glacial” in a very broad sense of “related to ice”. This is not uncommon but clearly inadequate in a differentiated treatment of various forms of perennial surface ice, and thereby causes unnecessary confusion. The paper makes perfectly clear that the central subject of the investigation, the ice aprons (IAs), are not “glaciers” (which are usually defined as surface ice showing clear signs of flow from an accumulation to an ablation area). The authors nevertheless in many places use formulations which disregard this essential fact. Already the title contains the expression “ice aprons … (and their) relations with other forms of glaciers” as if the ice aprons were a specific form of “glaciers” in contradiction to what is the clear conclusion of the study. The correct and precise term for all types of ice at the surface is “surface ice”; the term "perennial" can be added to make clear that such ice persists throughout the year. Within the general category “surface ice”, there are  “glaciers”, “glacierets”, “ice caps” (in the literature also called “miniature ice caps” as far as they cover peaks and crests in rugged mountain topography), “ice/snow covers” and “ice aprons”. The correct title of the corresponding overview (line 218, 4.2.) is therefore not “Glacier typology … and “glacier types” but should be “Typology of perennial surface ice and … for differentiating between them”. The authors must go through the entire manuscript and systematically eliminate the inconsistent wordings. On line 207, they correctly mention that it is beneficial to discriminate between glaciers and IAs - they must be consistent and strict about this. Examples of inconsistent wording are marked in the annotated PDF.

(b)  periglacial, permafrost and basic physics involved

As the authors correctly state, ice aprons are neither “glaciers” nor are they physically connected to them. They therefore do not belong to the “glacial” but to the “periglacial” realm. This periglacial belt “above glaciers” is sometimes called “upper periglacial belt” in contrast to the better known “lower periglacial belt” at altitudes below the equilibrium-line altitude on glaciers. This must be treated correctly. The connection and interaction between IAs and periglacial permafrost, on the other hand, is direct, evident and indeed by far the most important climate-related aspect of the investigated subject. The submitted paper provides a first statistical documentation of this fact but the relation has been recognized decades ago already and should be treated at an adequate state of knowledge. Already in the very first investigations of mountain permafrost in the Alps, ice aprons (in German: “Wandvereisung”) were understood to be related to bedrock permafrost and could, hence, be used as visible and widespread indicators of permafrost occurrence. The basis for this statement was an understanding of elementary physics involved. Like perennial ice patches on flatter ground, ice aprons are cold and frozen to their subsurface, because their ice cannot warm above 0°C in the warm season but cool down far below 0°C during wintertime. The resulting frozen condition stabilizes ice on steep slopes, because with temperate conditions, the ice would start sliding and could not remain in place on very steep slopes. At the same time, the frozen condition prevents subsurface rock destruction through freeze-thaw processes to take place. With the vanishing of ice aprons, therefore, the onset of such daily and seasonal freeze/thaw processes initiates active-layer formation, rock destruction and resulting rock-fall activity in the newly exposed rock slopes. Over extended time scales (decades to centuries), permafrost originally covered by cold ice aprons is likely to warm up to greater depths, progressively destabilizing larger masses of over-steepened bedrock. This phenomenon belongs to the serious climate-change impacts in icy mountains and should be adequately mentioned

The text could also note that falling rocks cannot stop on IAs. This is the reason why two important periglacial phenomena, IAs on slopes > 40° and talus on slopes < 35° are mutually exclusive.

Minor remarks

Examples of semantic inconsistencies to be improved and some other rather technical comments are marked in the annotated PDF.

I do not wish to remain anonymous as a reviewer. Some of the authors know me well and should feel free to contact me, if they need further information.

Author Response

Dear Professor Haeberli, we thank you for taking the time to read our paper and provide valuable inputs. 

Please find our replies to your comments in the attached pdf file.

Reviewer 2 Report

The authors presented a very complete and clearly written account of ice aprons and numerous glacier types on the Mont-Blanc Massif. The results are interesting, highlighting relationships between ice aprons, glaciers, topography, and setting. The new inventory and inferences made from analyzing this inventory, as well as the methods to conduct this study, are valuable contributions. I have only a few comments, and overall this was a pleasure to read.

- The introduction mentions that ice aprons are threatened by climate warming, but as they degrade do they have the potential to prime the landscape for landsliding or other forms of slope failure?

- This is outside of my expertise, but I found it hard to follow how the permafrost distribution fits in here. The consideration may be as simple as that if there is any ice cover then the ground beneath this is always below zero, so in the zone where this occurs due to air temperatures, the permafrost distribution is not affected by the presence of ice aprons. Or, perhaps could indicate that ice aprons are more likely to form where permafrost is expected? For glaciers, the base of the glacier can be warm (and wet), but that doesn’t necessarily mean that permafrost is degraded (or at least that could take time). I’m just trying to understand more of the significance of relating ice apron analyses to the permafrost distribution. Does the chance of developing or maintaining an ice apron change if it is underlain by permafrost? The rationale for bringing this in could be more clear when it is introduced around line 137, and potentially when it is discussed later in the manuscript.

- How thick may these ice aprons be?  (Sorry if I missed that)

- How generally applicable are inferences from MBM ice aprons? By using this population to provide a detailed definition, is there something to say about why this may hold for ice aprons in other locations? And, where else may ice aprons be found (are they possible anywhere there are cirque or slope glaciers in relatively steep topography)?

- This is also outside my direct expertise, but does the presence of a bergschrund require a headwall gap? Can that be imaged or accessed in a way to know if it does generate a cavity, or if the bergschrund is a crevasse that separate stagnant vs. moving ice at the headwall of the cirque glacier. The cartoons that I’ve seen are a bit different, but perhaps that configuration differs for cirque glaciers without headwall ice aprons?

Minor comments:

- I have also seen it come up this way for some reason, but Benn and Evans 2nd Edition was published in 2010 (not 2014)

- Line 35: If there is a word or two that can be added to clarify what “critical” refers to in this sentence that could add more weight since the introduction is at the beginning and it isn’t clear to the reader yet why these would be critical

- Line 78: “…low implications on water resources and their non-dynamic state” – this could be subtly rewritten for clarity; what is the implication of being stagnant?

- Line 109: I may have missed it, but was MBM defined?

- In general, there are many acronyms. I understand wanting to acronym ice apron since it is used so often, but being only two words that are both not very long, unless it is commonplace in the literature to use IA, I would suggest just using “ice apron” everywhere for readability. Either way would be fine, but that is my suggestion to consider.

- Could be worth mentioning somewhere in the main text that the mapping inventory is available for download (that is great!). Perhaps around Section 4.1

- Figure 8 axes labels are small. It took me a long time to understand what was being plotted, but reading the text it made the point simply. What is the value of the plot and is there a way to add legend labels or more information in the caption that would make this more clear to the reader?

- Figure 9 plot text looked a bit blurry to me. Some of the numbers were very hard to see.

- Could consider to bullet the key takeawyas in the conclusions, especially if it is anticipated that these apply widely to ice aprons elsewhere

- Last line of the conclusions could be a place to emphasize why these are critical

Author Response

Dear reviewer, we thank you for taking the time to read our paper and provide valuable inputs. 

Please find our replies to your comments in the attached pdf file.

Reviewer 3 Report

This paper describes the compilation and the evaluation of a comprehensive inventory of ice aprons (IAs) in the Mont Blanc Massif. The topic of ice aprons, very small and steep glaciers in rock faces, is underrepresented in glaciology and no regional-scale inventory of these particular features of the cryosphere is available to date. This paper fills in this gap based on the detailed mapping and analysis of high-resolution imagery. The manuscript is well written in general, and the approaches and interpretations are clear.  Nevertheless, a few aspects remain to be resolved as detailed below. Most importantly, the classification of all glaciers in the Mont Blanc Massif (Fig. 7 and Table 2) needs to be revised. This is only a side-aspect of the paper that is focused on ice aprons but requires quite some additional work to be pertinent. Just for this reason I request major revisions, although otherwise minor revisions would be sufficient to make this study acceptable.

Substantive comments:

-          Classification of glaciers: I agree that the classification performed here can give interesting insights. However, Figure 7 clearly shows that there is an important conceptual problem: A classification cannot cut a glacier in two parts! E.g.  Mer de Glace that is classified as a cirque glacier in the accumulation area and as a valley glacier in the ablation area. It never was the goal of any proposed glacier classification scheme to attribute individual regions of a single, i.e. dynamically connected glacier, to different classes. The separation becomes extremely subjective and you would be able to infinitely split each glacier into sub-regions, referring to individual types. Also, the classification appears highly subjective also in other cases. Eight small glaciers have for example been classified as Ice Caps. I do not think that according to the original definition or perception of ice caps, one would find such features in the Mont Blanc Area. I would suggest to completely omit this overall classification of glaciers. In fact, the paper is about ice aprons and does a very good job analyzing this new inventory. The glacier classification is thus just a side-aspect and is not needed for the conclusions presented here. If the authors decide to keep this part, I ask them to fully revise it and to classify only dynamically connected glaciers without splitting them.

-          Dating of the inventory / merging of different sources: The sources of the imagery are clearly laid out, but it did not become completely clear to me which data source (spread over several decades) was actually used for mapping the ice aprons for the final product, and thus to attribute a date for the feature. This would be essential in my opinion.

-          Importance of ice aprons: What is the relevance of ice aprons? The introduction mentions climbing routes through steep faces. However, this seems to be a relatively limited relevance, i.e. only important to a few mountaineers. Are there some other relevant aspects? E.g. rockfall hazard, dating of old ice (climate history), etc?

-          A general remark: For Switzerland and Austria recent and highly complete glacier inventories are also available. Why not try and detect IAs there using a quite straight-forward application of the definition provided? That would extend the scope of the study to the entire Alps. Well, maybe for a next paper.

Detailed comments:

-          Legend of Figure 2 (and text): Please refer to the actual data and publication shown here for GLIMS and the RGI. For the RGI, these outlines come from Paul et al 2011 and refer to the year 2003. I’m not entirely sure what the basis for the GLIMS outlines in the region is but this can certainly be looked up.

-          Related to the above: No, this is not the year these inventories refer to. It is 2003 for the RGI. For GLIMS, I assume that the latest French glacier inventory (Gardent et al., 2014, GPC) is included and it would be very important to reference this work and discuss the differences to the present inventory. Also the actual date of the outlines included in that inventory should be stated.

-          Figure 3: This figure makes clear an important aspect of the present work that deserves more discussion: ice aprons seem to be considered also as a part of larger glaciers, above the bergschrund. This seems to be a justified choice as ice flow across the bergschrund is limited or absent but it needs to be carefully laid out as conventional inventories would not cut contiguous ice masses apart. Furthermore, how error-prone is it to draw the lower boundary of the ice apron, i.e. can the bergschrund be detected well enough on all imagery used?

-          Figure 6: This figure could be condensed. At the moment there is a lot of free space.

-          Line 504: I find it very interesting that IAs almost do not occur on North facing slopes. Why? Please add some more discussion on this.

-          Figure 9: Why does the y-axis state A_T/A_MBM? What is A_T? I thought it was the area of ice aprons. The caption should give a quick explanation.

-          Line 520: It might be helpful to quickly repeat the meaning of the types here.

-          Line 606: I guess the limited snow accumulation explains why IAs are generally located far above the regional equilibrium line. I do however not agree with the statement in the following sentence that limited snow accumulation explains their retreat. Only a CHANGE in snow accumulation could lead to an imbalance. With climate change I would however rather expect more snow accumulation on IAs as temperatures during snow fall rise. The effect of increasing summer temperatures on melt rates is likely dominant.

-          Line 673: This is all clear and well written. However, I would have expected at least a few words on ice thickness, mass balance and the general climate response of IAs. That would complete the comprehensive characterization of these cryosphere features presented in this paper.

Author Response

(The authors gave the same response as above.)

Round 2

Reviewer 1 Report

The authors did a fine job with the revision. I just recommend to adjust the following two formulations

Line 80: Better “… does not significantly contribute …” (they indeed contribute but very little)

Line 665: Better: “… their ice temperature cannot exceed 0°C in summer…” (ice temperature never exceeds 0°C - this is basic physics)

With that, the interesting paper is ready for publication. I congratulate the authors for their excellent work.

Author Response

Dear Reviewer, 

Thanks a lot for your encouraging response and kind words of encouragement concerning our manuscript. We have incorporated the two minor suggestions you have in the revised draft. 

We thank you again for taking the time to review our manuscript in detail and providing valuable suggestions.